# Yellow Pigment Powders Based on Lead and Antimony: Particle Size and Colour Hue

**DOI:** 10.3390/jimaging7080127

**Published:** 2021-07-30

**Authors:** Giuseppe Capobianco, Giorgia Agresti, Giuseppe Bonifazi, Silvia Serranti, Claudia Pelosi

**Affiliations:** 1Department of Chemical Engineering Materials & Environment, Sapienza University of Rome, Via Eudossiana 18, 00184 Rome, Italy; giuseppe.capobianco@uniroma1.it (G.C.); giuseppe.bonifazi@uniroma1.it (G.B.); silvia.serranti@uniroma1.it (S.S.); 2Laboratory of Diagnostics and Materials Science, Department of Economics, Engineering, Society and Business Organization, University of Tuscia, Largo dell’Università, 01100 Viterbo, Italy; agresti@unitus.it

**Keywords:** artificial yellow pigments, particle size analysis, colour measurements, principal component analysis

## Abstract

This paper reports the results of particle size analysis and colour measurements concerning yellow powders, synthesised in our laboratories according to ancient recipes aiming at producing pigments for paintings, ceramics, and glasses. These pigments are based on lead and antimony as chemical elements, that, combined in different proportions and fired at different temperatures, times, and with various additives, gave materials of yellow colours, changing in hues and particle size. Artificial yellow pigments, based on lead and antimony, have been widely studied, but no specific investigation on particle size distribution and its correlation to colour hue has been performed before. In order to evaluate the particle size distribution, segmentation of sample data has been performed using the MATLAB software environment. The extracted parameters were examined by principal component analysis (PCA) in order to detect differences and analogies between samples on the base of those parameters. Principal component analysis was also applied to colour data acquired by a reflectance spectrophotometer in the visible range according to the CIELAB colour space. Within the two examined groups, i.e., yellows containing NaCl and those containing K-tartrate, differences have been found between samples and also between different areas of the same powder indicating the inhomogeneity of the synthesised pigments. On the other hand, colour data showed homogeneity within each yellow sample and clear differences between the different powders. The comparison of results demonstrates the potentiality of the particle segmentation and analysis in the study of morphology and distribution of pigment powders produced artificially, allowing the characterisation of the lead and antimony-based pigments through micro-image analysis and colour measurements combined with a multivariate approach.

## 1. Introduction

The preservation, archival, and study of cultural heritage is of the utmost importance at local, national, and international levels [1]. In the last decade, researchers in the field of imaging science have contributed to a growing set of tools for cultural heritage, thereby providing indispensable support to the above said efforts [2,3,4,5,6,7]. In this scenario, the morphological and morphometric analysis of the particles of pigments can supply a useful contribution to the knowledge and conservation of artistic objects. In general, the relationship between colour and particle size is known, and it is the object of studies in several fields such as food [8], earth science [9], and medicine [10]. The correlation between colour and particle size of artist pigments is also relevant, especially for the conservation and restoration of polychrome artifacts [11]. The knowledge of the optical characteristics of pigments used by artists and suppliers in earlier times represents an important starting point for the study and characterisation of paintings or generally of artworks, and for fixing the provenance of materials and techniques [12,13]. In fact, unlike the modern ones, the pigments used in the past were composed of particles different not in composition but also in size, morphological, and morphometric attributes. The optical properties of a pigment and in particular the hiding power, the tinting strength, and the colour depend on the dimensions and form of its grains. For example, pigments consisting of coarse grain particles commonly produce very saturated colour but have poor hiding power unlike those with fine grain, which instead have greater hiding power. In this study, a selected group of pigments were chosen to investigate for the first time the particle size and its influence on colour, and in particular, yellow pigments based on lead and antimony produced and used since ancient times according to different recipes and usually known as Naples yellow [14,15]. Previous studies showed that different pigments were produced by varying temperatures, times of firing, molar ratios of the reagents, kinds of crucible, and addition of melts and salts [16]. In order to evaluate the influence of particle size distribution on colour characteristics, starting from our previous studies, a group of yellow pigments produced in our laboratories from Pb and Sb elements or compounds were selected in the form of powders. These pigments were previously characterised through XRF (X-ray fluorescence) spectroscopy, SEM-EDS (scanning electron microscopy coupled with energy dispersive spectroscopy) analysis, and micro-Raman spectroscopy [14,15,16,17]. In particular, two groups of pigments, based on lead and antimony, were chosen for the present study: one group composed of three samples, named APB1, APB2, and APB3, produced according to the recipe by Valerio Mariani from Pesaro with the addition of NaCl, and another group including two samples, named PSAPPB1 and PSAPPB2, synthesised according to the common recipe of Cipriano Piccolpasso and Giambattista Passeri [18,19]. The ancient recipes did not report the firing temperature for producing artificial yellow pigments, so different values were tested for evaluating the influence of temperature on morphology, colour, and compositions of the powders [17]. In spite of a considerable number of articles and studies on these yellow pigments, some questions still remain partly solved [20,21,22]. In particular, the reasons why different chemical compounds resulted starting from the same reagents are not completely clear, as well as the alternate use of these yellows since the Middle Ages for polychrome artifacts [17,19]. However, some studies showed that the use of a particular type of yellow could be indicative of a specific historical period and geographical area in some cases, such as those reported by Montanari et al., resulting in a complete re-assessment of production centres and modalities [12,13]. Thus, the study of artificial yellow pigments is relevant for gathering information about the possible historical period of artworks, the geographical area of production or provenance, and potentially for attribution purposes.

This study aims to investigate the morphological and morphometrical parameters of Pb/Sb yellow pigment particles and to link them to the chromatic characteristics. The specific objective of the work is to combine the results obtained through reflectance spectrophotometry in the visible range, with the morphological and morphometric parameters obtainable by stereo microscope acquisition. All data were elaborated by statistical and chemometric tools in order to verify the significance of the measurements and to compare all variables analysed at the same time.

## 2. Materials and Methods

### 2.1. Samples’ Description and Image Acquisition

Table 1 lists the pigment powders used in this study, the modality of synthesis, and the old recipes from which they were obtained. PSAPPB2 was obtained starting from PSAPPB1 with an annealing process carried out under the same conditions used for producing PSAPPB1, according to the procedure reported in the recipes indicated in Table 1.

In the recipes by Cipriano Piccolpasso and Giambattista Passeri, the amounts of reagents are reported in Roman *libra* (lb, lire in the recipes), specifically: Sb 4 lb, Pb 6 lb, feccia (lees) 1 lb (1 lb corresponds to 327.168 g).

For the production of lead–antimony-based yellow pigments, pure grade chemicals supplied by Acros Organics (New York City, New York), MP Biomedicals (Santa Ana, California), and Sigma-Aldrich (St. Louis, Missouri) were used. The reagents were mixed in agate mortars by following the amounts suggested in the recipes and then placed into the laboratory furnace at room temperature. They were then heated in order to reach the required temperature. The temperature was maintained constant for 5 h. In the case of yellows prepared according to the common recipes by Cipriano Piccolpasso and Giambattista Passeri, a double firing was used [16].

### 2.2. Colour Measurements

Colour was measured through an X-Rite CA22 reflectance spectrophotometer according to the CIELAB colour system [23]. The characteristics of the colour measuring instrument are the following: light source D65; standard observer 10°; fixed geometry of measurement 45°/0°; spectral range 400–700 nm; spectral resolution 10 nm; aperture size 4 mm. For each specimen, twenty-five measurements were performed in order to account for possible colour variations due to particle size of powders. Samples were mixed after each measure, and then the average values and standard deviations were calculated. Measurements were performed at room temperature (about 20 °C) and relative humidity of about 50%, controlled by the laboratory humidifier/dehumidifier.

### 2.3. Stereomicroscopic Investigation

Samples were characterised by optical microscopy using a Leica M205C stereomicroscope. A coaxial LED incident-light illumination optic unit was utilised as an energising source. The adopted magnification was 160× to obtain images of powder samples and details of the morphological and morphometric parameters. The same areas were acquired for each sample under transmitted light so that to highlight the morphological characteristics of the examined powders.

### 2.4. Data Processing

The process of segmentation has been performed using the MATLAB software environment (Version 7.11.1, MathWorks, Inc., Natick, MA, USA). The image segmentation is commonly used to process and analyse digital images with the aim of creating parts or regions, often on the base of the pixel characteristics. In order to maximise the segmentation of the elements in the image, the first step of processing is devoted to the separation of the background from the foreground and the grouping of pixel regions according to similarities in colour or forms. Therefore, algorithms are needed to transform the grey scale image into a binary image (binarisation), so as to preserve the relevant content as much as possible (Figure 1A). At the same time, all objects less than 100 pixels in size were removed. Subsequently the measures of each object in the binary image were extracted (Figure 1B).

The extracted parameters can be listed as follows:Area (Area): effective number of pixels in the region, returned as scalar.Centroid (Circ.): mass centre of the region, returned as a 1-by-Q vector. The first centroid element is the horizontal (or x coordinate) of the mass centre. The second element is the vertical coordinate (or y coordinate). The other elements of the centroid are ordered by size.Eccentricity (Ecc.): eccentricity is the ratio between the distance of the ellipse fires and the length of its major axis. The value ranges between 0 and 1.Major axis length (M axis): length (in pixels) of the major axis of the ellipse that has the same second normalised central moments of the region, returned as scalar.Minor axis length (m axis): length (in pixels) of the minor axis of the ellipse that has the same second normalised central moments of the region, returned as scalar.Equivalent diameter (Eq. diameter): Diameter of a circle with the same area of the region, returned as scalar. Calculated as sqrt (4*Area/pi).Perimeter (Perim.): the distance around the boundary of the returned region as a scalar. The system calculates the perimeter by measuring the distance between each pair of adjacent pixels around the edge of the region.Hausdorff Fractal (F. Haus.): returns the Hausdorff fractal dimension of an object represented by a binary image.Fractal “Box-Counting” (F. boxc.): counts the number N of D-dimensional boxes of size R necessary to cover the % of the non-zero elements of the identified object.

### 2.5. Principal Component Analysis (PCA)

Principal component analysis (PCA) is a powerful and versatile method capable of providing an overview of complex multivariate data. It is widely adopted to treat different kinds of data [24,25,26]. PCA can be used to reveal relations between variables and samples (i.e., clustering), detecting outliers, finding and quantifying patterns, generating new hypotheses, etc. PCA is used to decompose the data into several principal components (PCs), linear combinations of the original data, embedding the variations of each collected data set [24]. According to this approach, a reduced set of factors is produced. Such a set can be used for exploration, since it provides an accurate description of the entire dataset. The first few PCs, resulting from PCA, are generally used to analyse the common features among samples and their grouping: in fact, samples characterised by similar characteristics tend to aggregate in the score plot of the first two or three components [26].

Since the selected variables differ from each other, the samples were pre-processed through Autoscale. In this paper, PCA was applied to both imaging and colour values.

## 3. Results and Discussion

### 3.1. Image Analysis

The images of the three areas acquired under reflected and transmitted light are shown in Figure 2 and Figure 3.

From the microscopic images, it appears that samples are not homogeneous in particle size and distribution, and also in morphological aspect. After the process of segmentation, the obtained parameters, as described in Section 2.4, with the relative abbreviations, are reported in Table 2. The analysis of average data shows morphological analogies between samples APB1, ABP2, and APB3 with fractal dimensions comparable between the three samples. However, the morphological variability between the examined samples, highlighted by the minimum and maximum values, is high. Samples PSAPPB1 and PSAPPB2 exhibit significant differences mainly in the contour variations. This is stressed by the different fractal dimension and by a decrease of dimension highlighted by the following parameters: axis, perimeter, and average equivalent diameter.

The high variance found in all samples requests the use of multivariate methods for the analysis variance. The PCA model of APB1, APB2, and APB3 requires six PCs to express a total captured variance equal to 99.35% and shows a complex clusters scenario. In fact, the score clusters of the three classes are not sharply separated by a single PC, except for the ‘APB3′ class. In more detail, the PC1-PC5 score plot (Figure 4A) shows that pixels belonging to ‘APB1′ and ‘APB2′ classes occur in different regions of the plot in respect to APB3 and they are not separated. In addition, APB1 and APB2 are clustered in two different portions of the score plot, probably due to the major variability of particles in terms of grain size and morphology. By analysing the loadings of the selected parameters, it is possible to highlight how PC5, that mostly influences the variance detected between APB3 and the other two samples (APB2 and APB1), is mainly due to the two fractal parameters, while morphologically the samples are similar to each other as shown by PC1 (Figure 4B).

The variation found in the APB1 and APB2 samples is given by the presence of areas with different circularity as shown by PC1. This result is in agreement with the previous published studies on artificial yellow pigments that showed differences in composition and colour between the three powders produced according to the recipe by Valerio Mariani from Pesaro (1620) [18]. APB1 and APB2, in fact, were found to be inhomogeneous powders with yellow and brown grains, especially APB1, whose composition was not exactly characterised also by applying X-ray diffraction (XRD) analysis, as discussed in [18] and in Appendix A included in the present paper.

It has been supposed that different compounds were produced also containing Na and Cl in the crystalline lattice of lead antimonate [18]. On the other hand, sample APB3 was homogeneous in colour and composition, in accordance with the results of particle analysis.

The PCA model of PSAPP1 and PSAPP2 requires seven PCs to express a total captured variance equal to 99.89%. The score clusters of the two classes are well separated by a single PC. In detail, the PC1-PC5 score plot (Figure 5A) shows that pixels belonging to ‘PSAPP1′ and ‘PSAPP2′ classes occur in different regions of the plot. In addition, PSAPP2 pixels are clustered in two different portions of the score plot, probably due to the presence of particles with different grain sizes and morphology. By analysing the loadings plot, it can be derived how PC5, mostly influencing the variance between the PSAPP1 and PSAPP2, is mainly determined by the two fractal parameters (Figure 5B). On the other hand, the samples are morphologically similar to each other, as shown by PC1. The variability observed within the yellow PSAPPB2 is caused by areas of different circularity, as suggested by the loading values of PC1.

### 3.2. Colorimetric Analysis

The other important parameter considered in the present paper is the colour of the produced powders. Hue of painting pigments is highly relevant in the choice of materials by artists that probably knew the production modalities and the different kinds of available yellows [19].

The average values of the chromatic coordinates with the relative standard deviation are reported in Table 3. Comparing the values of chromatic coordinates for the three yellows prepared according to the recipe of the treatise by Valerio Mariani from Pesaro, i.e., APB1, APB2, and APB3, we observe a clear difference of APB3 with respect to APB1 and APB2, especially concerning the a* coordinate that is higher in APB3 indicating a reddish hue of the pigment. APB3 is also darker in respect to APB1 and APB2 and more yellow, with the b* coordinate having a higher value.

The other two yellows (PSAPPB1 and PSAPPB2) exhibit very similar values of the b* coordinate, representing the yellow component, and also a similar value of a*. The most consistent difference between the two yellows is given by the L * parameter: the lower value of about 4 points indicates that the PSAPPB2 sample is darker than PSAPPB1. Therefore, the annealing process does not seem to change the value of the chromatic coordinates but only causes a moderate darkening of the pigment. Furthermore, the reflectance spectra of samples APB1, APB2, and APB3, as collected and after pre-processing (Figure 6A,B), have been compared to highlight the variations in the different spectral regions. The spectra of APB1, APB2, and APB3 (Figure 6A) show variations at 500 and 600 nm, whereas those of PSAPPB1 and PSAPPB2 exhibit an increasing trend from 400 to 650 nm, with a slight variation after 650 nm.

In order to maximise the spectral differences, MSC (Median) pre-processing and 1st Derivative have been applied with the aim of removing the light scattering effects on the pigment surface and of emphasising the variations of the spectral signatures, respectively. Finally, PCA was applied to the pre-processed spectra (Figure 7).

The scores plot of PCA (Figure 7A) shows five clusters corresponding to the five yellow pigments based on lead and antimony. In more detail, it is possible to note that spectra belonging to the ‘APB3’ class can be easily distinguished from pixels of the other classes, being clustered in the fourth quadrant of PC1-PC2. In addition, the PCA score plot shows that pixels belonging to the ‘APB1′ and ‘APB2′ classes are mainly concentrated in the first quadrant, corresponding to positive values of PC1 and PC2, whereas pixels belonging to ‘PSAPPB1’ and ‘PSAPPB2’ classes occur in different regions of the plot, mainly in the second and third quadrants, corresponding to negative values of PC1, and they are very close to each other. Moreover, the loading plot associated with PCA (Figure 7B) highlights how the variance of positive PC1 is mainly related to the wavelength around 550 nm, whereas the negative PC1 is influenced by the spectral region around 450 and 600 nm. The positive variance of PC2 is influenced by the wavelength around 500 nm, whereas the negative variance of PC2 is influenced by the spectral region around 550 nm.

### 3.3. Comparison of Colour and Particle Analysis

Comparing the results of particle and colour analysis with the previous published data on lead and antimony-based yellow pigments, an agreement can be assessed. PSAPPB1 and PSAPPB2, in fact, have been found constituted by inhomogeneous powders with different stoichiometric ratios of Pb and Sb in the areas examined under SEM-EDS [16]. These two yellows include two main lead antimonates, i.e., Pb_2_Sb_2_O_7_ and PbSb_2_O_6_, the last being rosiaite often found in the synthesis of Naples yellow [14,27], but also other compounds containing K are not well-characterised. K has been detected through SEM-EDS analysis in all examined points [14].

The results of colour measurements on PSAPPB1 and PSAPPB2 show little variation between the two samples if compared to the variations detected by particle analysis demonstrating how the annealing process carried out on PSAPPB1 for obtaining PSAPPB2 decreases the uniformity of powder in terms of particle size, without losing the acquired colorimetric features, albeit with a small decrease in brightness. This result is interesting from a technological point of view because it suggests that probably a second firing was not necessary as it decreases the particle homogeneity without significantly changing the colour characteristics.

The particle and colour analysis of the APB1 and APB2 samples confirms that application of slightly different temperatures (900 °C for APB1, and 950 °C for APB2) does not produce significant differences between the two samples. On the other hand, the APB3 sample, fired at a temperature of 1050 °C, has completely different characteristics in respect to APB1 and APB2, both in terms of particle and colour parameters. Specifically, the colour data of APB1, APB2, and APB3 samples show how APB1 and APB2 have similar colour features if compared to APB3; moreover, the first two yellows exhibit particle heterogeneity in respect to APB3, confirming the results obtained by previous findings on chemical composition [16]. In fact, APB1 and APB2 are characterised by a compositional heterogeneity, while APB3 is a homogeneous dark yellow powder with a reddish hue, as shown by the significantly higher value of the a* coordinate.

## 4. Conclusions

A new approach has been employed in this study to investigate powder samples of artificial yellow pigments, based on lead and antimony as main elements, widely used since ancient times for ceramics, glasses, and paintings. These pigments are very important in the study of artworks because their composition is linked to ancient recipes that could allow us to suppose the geographical areas of their provenance or the potential circulations of materials and techniques, as recently demonstrated by Montanari et al. [12,13].

Yellow pigments based on Pb and Sb were produced with different recipes with the addition of salt (NaCl) and/or K-tartrate, but temperature, times, and crucible types were not specified in the recipes, so different experimental tests have been performed producing pigments of similar colour and appearance but of different composition. Moreover, in the case of Naples yellow, different compounds were obtained in the firing process, independently from the starting reagents. This results in inhomogeneous compositions of Naples yellow prepared according to ancient recipes.

The analysis proposed in the present paper, being non-invasive and rapid, resulted in being useful for the examination of pigment powders by giving information about morphology, distribution, and homogeneity. In detail, the optical and colorimetric characteristics of yellow pigments are correlated with particle sizes investigated by image analysis combined with a multivariate approach.

The results obtained on the yellow powders are interesting in terms of pigment particle homogeneity and colour, demonstrating that pigments having uniform colour are not always also characterised by uniform particle parameters. In general, the pigments produced according to the ancient recipes are not homogeneous in particle characteristics (i.e., size and shape) and composition, even if the colour is homogeneous. Particle analysis demonstrated the importance of this approach to evaluate the analogies and differences between yellows in terms of morphologic and morphometric parameters, and in particular for dimension (i.e., area, circularity, perimeter, equivalent diameter) and fractal dimension.

In summary, the proposed approach makes it possible to obtain a great quantity of information on pigments through a non-destructive approach, allowing us to carry out subsequent complementary analyses on the same samples that could complete the information extractable from the pigment powders and from the production processes.

## Figures and Tables

**Figure 1 jimaging-07-00127-f001:**
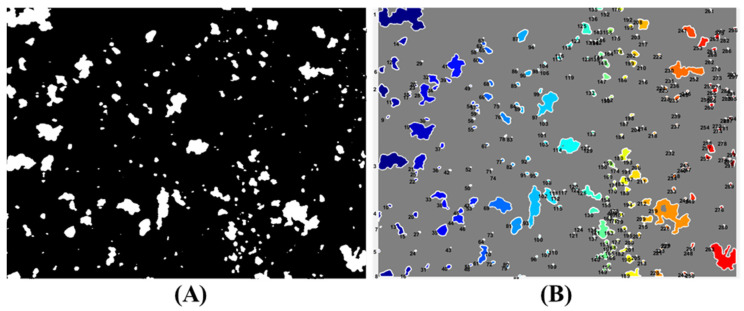
Sample APB1: binary image (**A**) and labelled domains (i.e., objects) (**B**) whose morphological and morphometric attributes were computed.

**Figure 2 jimaging-07-00127-f002:**
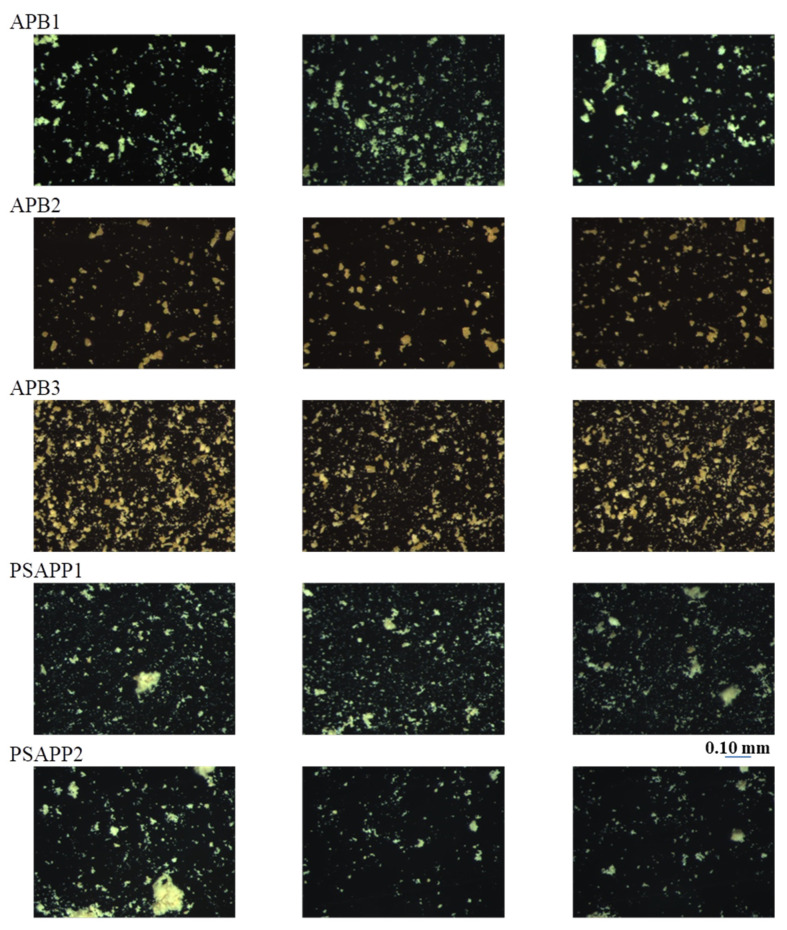
Stereo microscope photomicrographs of the three selected areas for each sample at 160× magnification, under reflected light.

**Figure 3 jimaging-07-00127-f003:**
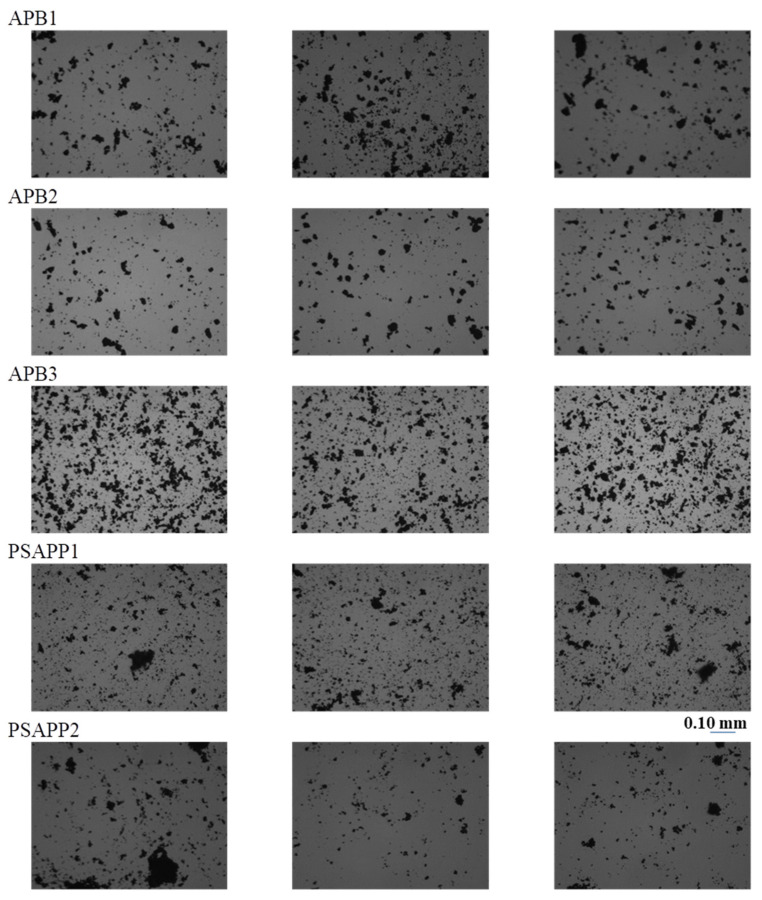
Photomicrographs of the three selected areas for each sample, under transmitted light at 160× magnification.

**Figure 4 jimaging-07-00127-f004:**
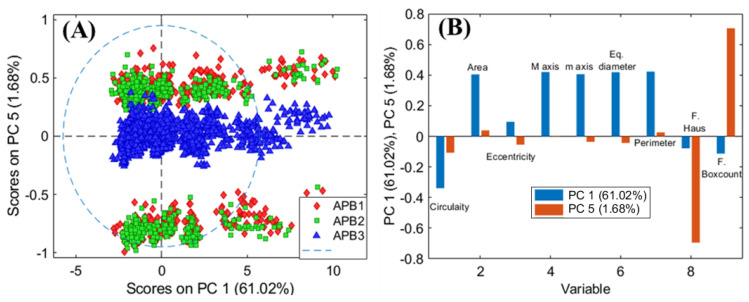
PCA score plot of PC1 and PC5 (**A**), and loadings (**B**) showing the variability of each selected parameter for the samples APB1, APB2, and APB3.

**Figure 5 jimaging-07-00127-f005:**
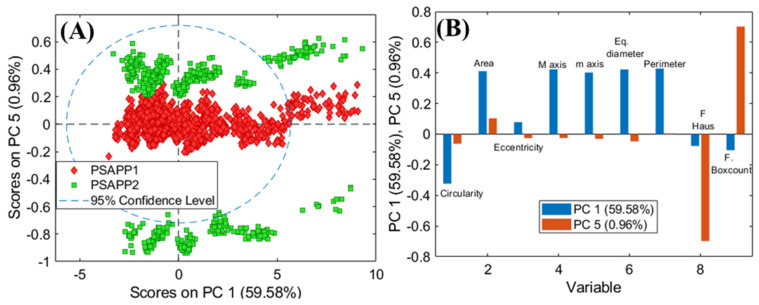
PCA score plot of PC1 and PC5 (**A**), and loadings (**B**) showing the variability of each selected parameter for the samples PSAPPB1 and PSAPPB2.

**Figure 6 jimaging-07-00127-f006:**
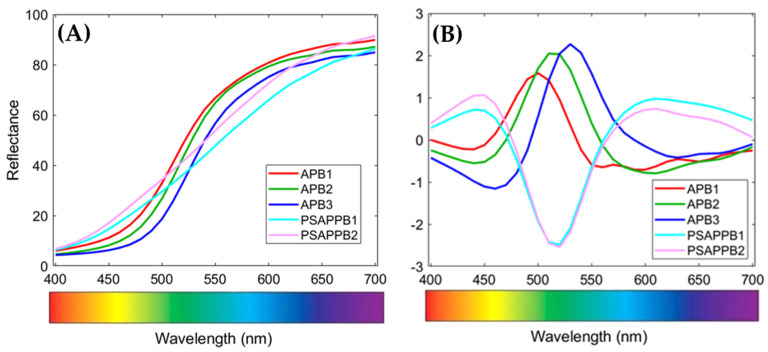
Reflectance spectra of artificial Pb/Sb yellows (**A**) and pre-processed spectra (**B**) through MSC (Median), 1st Derivative (order: 2, window: 5 pt, including only tails: weighted), Mean Centre.

**Figure 7 jimaging-07-00127-f007:**
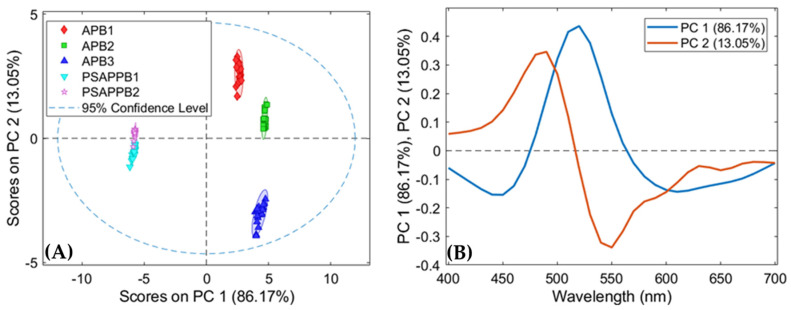
Score plot (**A**) of principal component analysis and loadings (**B**) of the reflectance data of the yellow pigments.

**Table 1 jimaging-07-00127-t001:** List of pigments used for particle size and colour investigation and synthesis details.

Abbreviation	Reagents	Reagent Weight (g)	Experimental Conditions	Historical Recipe
APB1	PbO, Sb_2_O_3_, NaCl	1.42, 0.85, 0.57	t = 900 °C for 5 h porcelain crucible	Trattato di Valerio Mariani da Pesaro (1620), *Giallo dei vasari* [21]
APB2	PbO, Sb_2_O_3_, NaCl	1.42, 0.85, 0.57	t = 950 °C for 5 h porcelain crucible	Trattato di Valerio Mariani da Pesaro (1620), *Giallo dei vasari* [21]
APB3	PbO, Sb_2_O_3_, NaCl	1.42, 0.85, 0.57	t = 1050 °C for 5 h porcelain crucible	Trattato di Valerio Mariani da Pesaro (1620), *Giallo dei vasari* [21]
PSAPPB1	Sb_2_O_3_, PbO, C_4_H_5_KO_6_	1.30, 1.90, 0.30	t = 800 °C for 5 h on a terracotta tile	Cipriano Piccolpasso, *I Tre Libri dell’Arte del Vasaio, f. 29v8* (1559) and Giambattista Passeri, *Istoria delle Pitture in Majolica fatte in Pesaro* (1758) [17,19]
PSAPPB2	PSAPPB1	1.00	t = 800 °C for 5 h on a terracotta tile	Cipriano Piccolpasso, *I Tre Libri dell’Arte del Vasaio, f. 29v8* (1559) and Giambattista Passeri, *Istoria delle Pitture in Majolica fatte in Pesaro* (1758) [17,19]

**Table 2 jimaging-07-00127-t002:** Average values of the parameters identified for each class.

Sample	Circ.	Area	Ecc.	M Axis	m Axis	Eq. diameter	Perim.	F. Haus.	F. boxc.
**PSAPPB1**	0.94	63.00	0.82	12.07	6.90	8.96	29.07	1.09	1.16
max PSAPP1	1.00	73871.00	0.99	574.04	279.11	306.68	2182.01	1.30	1.33
min PSAPPB1	0.11	5.00	0.15	4.94	1.15	2.52	8.00	0.92	1.00
**PSAPPB2**	0.78	1422.29	0.71	40.49	24.90	29.95	120.80	1.01	1.08
max PSAPPB2	1.00	165626	0.98	694.43	426.77	459.22	2660.88	1.20	1.25
min PSAPPB2	0.14	5.00	0.21	5.63	1.15	2.52	8.00	0.92	1.00
**APB1**	0.77	1714.82	0.71	45.97	28.57	33.96	140.18	1.01	1.08
max APB1	1.00	47063.00	0.98	433.91	199.67	244.79	1658.16	1.20	1.25
min APB1	0.13	6.00	0.10	5.21	2.11	2.76	8.83	0.92	1.00
**APB2**	0.82	1388.81	0.70	40.94	25.85	31.15	118.32	1.02	1.08
max APB2	1.00	29496.00	0.98	408.31	184.30	193.79	1142.87	1.20	1.25
min APB2	0.27	8.00	0.10	6.22	2.02	3.19	10.83	0.92	1.00
**APB3**	0.76	1626.80	0.73	46.48	27.51	32.64	146.76	1.01	1.08
max APB3	1.00	71160.00	0.99	506.89	383.18	301.00	3611.16	1.20	1.25
min APB3	0.07	6.00	0.05	5.26	1.89	2.76	10.24	0.92	1.00

**Table 3 jimaging-07-00127-t003:** Average values and standard deviation of the chromatic coordinates of yellow samples.

Sample	L*	a*	b*
APB1	82.5 ± 0.91	7.41 ± 0.58	67.2 ± 0.46
APB2	80.9 ± 0.73	9.60 ± 0.47	73.6 ± 0.50
APB3	77.0 ± 0.69	15.5 ± 0.60	75.7 ± 0.67
PSAPPB1	74.9 ± 0.80	12.5 ± 0.39	48.1 ± 0.23
PSAPPB2	78.4 ± 0.40	11.3 ± 0.33	48.7 ± 0.38

## Data Availability

Data is contained within the article or Appendix A.

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
