# Peer review of "Yellow Pigment Powders Based on Lead and Antimony: Particle Size and Colour Hue"

_2313-433X, 2021, doi:10.3390/jimaging7080127_

Round 1
Reviewer 1 Report
The authors are well placed to attempt a study of particle size analysis and colour measurements of yellow powders that were synthesized in laboratory according to ancient recipes. Principal Component Analysis was applied in order to detect differences and analogies on the base of particle size distribution and colour.
Specific comments
Table 1: It is not clear what the difference between PSAPPB1 and PSAPPB2 is. One should proceed to the section 3.3 to understand that a longer annealing process was carried out on PSAPPB1.
It would be useful for the reader, if the authors indicate the abbreviations used for the parameters in Table 2 at the list of parameters presented from line 138 to line 157.
Line 274: “other compounds containing K and not well-characterized” should be replaced with “other compounds containing K are not well-characterized”.
I would recommend to the authors to present their XRD data (it is mentioned that XRD method was applied, line 214). It would be useful for the reader to understand whether there is a difference in chemical composition of the synthesized pigments.
Author Response
The authors are well placed to attempt a study of particle size analysis and colour measurements of yellow powders that were synthesized in laboratory according to ancient recipes. Principal Component Analysis was applied in order to detect differences and analogies on the base of particle size distribution and colour.
AUTHOR REPLY: thank you for the general comment to the paper
Specific comments
Table 1: It is not clear what the difference between PSAPPB1 and PSAPPB2 is. One should proceed to the section 3.3 to understand that a longer annealing process was carried out on PSAPPB1.
AUTHOR REPLY: thank you for the comment. We specified the difference between PSAPPB1 and PSAPPB2 at the beginning of sub-paragraph 2.1 in order to have immediately cleared such a difference
It would be useful for the reader, if the authors indicate the abbreviations used for the parameters in Table 2 at the list of parameters presented from line 138 to line 157.
AUTHOR REPLY: we agree with the reviewer. We added the abbreviation used in the Table 2, at the list reported previously
Line 274: “other compounds containing K and not well-characterized” should be replaced with “other compounds containing K are not well-characterized”.
AUTHOR REPLY: thanks for the suggestion. We corrected the sentence
I would recommend to the authors to present their XRD data (it is mentioned that XRD method was applied, line 214). It would be useful for the reader to understand whether there is a difference in chemical composition of the synthesized pigments.
AUTHOR REPLY: XRD data for samples APB1, APB2 and APB3 were published in the paper reported at ref [18]. However, as recommended by the reviewer, XRD results were added as Supplementary Materials.
Reviewer 2 Report
Excellent work indeed. You might describe the spectrophotometric limitations and the influence on the recorded spectra. The measurement conditions depend on the compaction level of the powders and of air moisture. Though they have a generally weak influence on the global aspect of the reflectance spectrum (modification in magnitude) the chosen device is not contact free with the sample to be measured. It could also interesting to use a contactless device to make further comparisons with pigments participating to an artwork.
A very weak amount of spelling problems could be corrected while they are not essential for a well comprehensive reading.
Author Response
Excellent work indeed. You might describe the spectrophotometric limitations and the influence on the recorded spectra. The measurement conditions depend on the compaction level of the powders and of air moisture. Though they have a generally weak influence on the global aspect of the reflectance spectrum (modification in magnitude) the chosen device is not contact free with the sample to be measured. It could also interesting to use a contactless device to make further comparisons with pigments participating to an artwork.
AUTHOR REPLY: thank you so much for appreciating our paper. Yes, the reflectance spectrophotometer that we used works in contact modality and it could be useful to have a contactless device especially for measuring real work of art. For the present study we would like to focus the attention on particle analysis and to it correlation to colour, but certainly a next step will be addressed to apply contactless device to measure pigments both in powder and applied on mock-ups, and further on artworks.
In order to limit the variability of reflectance spectra, due to the manual grinding and measurements, we chose to perform several acquisitions on a same sample in order to calculate average values and standard deviation.
To better understand the colour data, also in terms of average values and standard deviation, we added a Table (Table 3) with the values of L*a*b* coordinates and the relative standard deviation. We also added discussion on this.
We also added information about the measurements’ environmental conditions (our laboratory).
A very weak amount of spelling problems could be corrected while they are not essential for a well comprehensive reading.
AUTHOR REPLY: thank you for the suggestion. We carefully checked the manuscript for eliminating the spelling problems. Corrections have been highlighted by red characters in the manuscript.